# In Silico Study of piRNA Interactions with the SARS-CoV-2 Genome

**DOI:** 10.3390/ijms23179919

**Published:** 2022-08-31

**Authors:** Aigul Akimniyazova, Oxana Yurikova, Anna Pyrkova, Aizhan Rakhmetullina, Togzhan Niyazova, Alma-Gul Ryskulova, Anatoliy Ivashchenko

**Affiliations:** 1Higher School of Medicine, Faculty of Medicine and Healthcare, Al-Farabi Kazakh National University, Almaty 050040, Kazakhstan; 2Faculty of Biology and Biotechnology, Al-Farabi Kazakh National University, Almaty 050040, Kazakhstan; 3Center for Bioinformatics and Nanomedicine, Almaty 050060, Kazakhstan; 4Institute of Biochemistry and Biophysics, Polish Academy of Sciences, 02-106 Warsaw, Poland; 5Department of Population Health and Social Sciences, Kazakhstan’s Medical University “KSPH”, Almaty 050060, Kazakhstan

**Keywords:** SARS-CoV-2, COVID-19, coronavirus genome, piRNA, miRNA, drug target

## Abstract

A prolonged pandemic with numerous human casualties requires a rapid search for means to control the various strains of SARS-CoV-2. Since only part of the human population is affected by coronaviruses, there are probably endogenous compounds preventing the spread of these viral pathogens. It has been shown that piRNA (PIWI-interacting RNAs) interact with the mRNA of human genes and can block protein synthesis at the stage of translation. Estimated the effects of piRNA on SARS-CoV-2 genomic RNA (gRNA) in silico. A cluster of 13 piRNA binding sites (BS) in the SARS-CoV-2 gRNA region encoding the oligopeptide was identified. The second cluster of BSs 39 piRNAs also encodes the oligopeptide. The third cluster of 24 piRNA BS encodes the oligopeptide. Twelve piRNAs were identified that strongly interact with the gRNA. Based on the identified functionally important endogenous piRNAs, synthetic piRNAs (spiRNAs) are proposed that will suppress the multiplication of the coronavirus even more strongly. These spiRNAs and selected endogenous piRNAs have little effect on human 17494 protein-coding genes, indicating a low probability of side effects. The piRNA and spiRNA selection methodology created for the control of SARS-CoV-2 (NC_045512.2) can be used to control all strains of SARS-CoV-2.

## 1. Introduction

At the beginning of the pandemic, the effect of miRNA (mRNA inhibiting RNA) on the protein expression of the SARS-CoV-2, encoding genes was predicted [1,2]. Therefore, an attempt was made to identify miRNAs capable of binding to whole coronavirus gRNA (genome RNA) and suppressing its reproduction [3]. The described method of coronavirus control was to identify miRNAs capable of efficiently binding to coronavirus gRNA [1,2]. To increase the efficiency of miRNAs, synthetic miRNAs (smiRNA) based on them were designed, which were able to bind much more strongly than endogenous miRNAs. Attempts to create smiRNA based on endogenous human miRNAs or engineered siRNA (short interfering RNA) to control COVID-19 were successful in experiments on cell culture and experimental animals [3]. SARS-CoV-2 multiplication in human cell culture and hamsters was suppressed 99 to 100% by smiRNA administration. However, clinical trials revealed side effects of the miRNA-based drugs that served as a limitation for their use in the treatment of COVID-19 [3,4,5]. The reason for these results was probably the side effect of smiRNA and siRNA on human protein-coding genes. An evaluation of the effect of these smiRNAs and siRNAs on human genes has been proposed in our publications [1,2]. Finding effective smiRNAs and siRNAs in the fight against SARS-CoV-2 without first evaluating their side effects could prolong the fight against coronaviruses for years to come. As a result of the conducted computational studies on the effect of miRNAs and piRNAs (PiWi-interacting RNAs), the direct effect of piRNAs on the protein synthesis of the coronavirus was established for the first time, which may open up new possibilities for fighting the COVID-19 pandemic.

We started a study on the possible effect of piRNA molecules on SARS-CoV-2 (NC_045512.2) replication. The piRNAs are on average eight nucleotides longer than the miRNAs and can bind more strongly to the genomic RNA (gRNA) of the coronavirus [6,7]. Unfortunately, notions about the biological role of piRNAs remain insufficiently substantiated in the many years since their discovery [7]. The putative involvement of piRNAs in the regulation of transposon movement control requires direct evidence in addition to correlations [8,9]. There are reports on the involvement of piRNAs in the regulation of gene expression, but these reports are conjectures without specifying a specific mechanism for this process [10,11,12,13,14]. Attempts to detect the effect of piRNA on coronaviruses were reviewed recently in [15]. It has been shown that piRNAs in exosomes affect SARS-CoV-2 reproduction. piRNAs involving PIWI alone can suppress the SARS-CoV-2 virus. Unfortunately, studying the role of piRNAs in suppressing coronavirus multiplication will be long without establishing to what extent each piRNA can interact with the gRNA of coronavirus strains [16]. Some properties of piRNA are similar to those of miRNAs, so the binding of piRNA to the coronavirus genome has been studied [16,17]. Given the lack of significant advances in the study of the biological role of a few thousand miRNAs, scientists have great uncertainty about the study of more than eight million human piRNAs contained in the piRNA database [10,12].

## 2. Results

In the present work, we show the possible involvement of piRNAs and spiRNAs (synthetic miRNAs) in the regulation of protein synthesis in the gRNA of the SARS-CoV-2 coronavirus. Since translation occurs during virus multiplication, including SARS-CoV-2, it is logical to investigate the interaction of piRNAs with the RNA of this coronavirus genome. This problem is currently relevant for developing methods to combat the pandemic caused by SARS-CoV-2. Our study demonstrates the potential application of piRNAs in the treatment of SARS-CoV-2-induced disease. We can evaluate in silico the side effects of synthetic piRNAs on human genes, and with this evaluation in mind, this work proposes piRNAs and spiRNAs for use as drugs against SARS-CoV-2.

In the introduction, we hypothesized that piRNAs can regulate RNA translation, such as mRNA protein synthesis, and single-stranded RNA replication using gRNA viruses as examples. To test this assumption, we analyzed the binding of more than eight million piRNAs to SARS-CoV-2 gRNA and found that several dozen piRNAs can bind to coronavirus gRNA. The pathogenicity of the virus when its gRNA proliferates intensively in the cell may manifest in the binding of piRNAs that are essential for cell life, which would lead to increased expression of piRNA target genes and disrupted metabolism, including the development of pathologies. Without considering this option, we examined the binding of piRNA to the gRNA of the virus. The BSs of piRNAs are found along the entire length of the gRNA of the coronavirus and have important features. We identified several piRNA BSs organized into BS clusters representing gRNA site binding up to several dozen piRNAs with overlapping nucleotide sequences.

### 2.1. Thirteen piRNA Binding Cluster in SARS-CoV-2 gRNA

The first cluster of human endogenous 13 piRNA BSs is localized from 7176 nt (piR-177814) to 7178 nt (piR-158266) with a length of 28 nt (Figure 1). The nucleotides in these clusters encode the oligopeptide in the ORF1ab protein. The identified piRNAs interact weakly with gRNA. The average value of the free energy of interaction between piRNA and RNA is −102 ± −4 kJ/mol, but a large amount of piRNA ensures the suppression of viral protein synthesis and viral gRNA replication. The action of endogenous piRNAs on target genes depends, according to the laws of dynamic biochemistry, on the ratio of piRNA and mRNA concentrations of protein-coding genes. Only such large sets of piRNAs binding to coronavirus gRNAs in the BS gRNA coronavirus cluster could protect the organism from these pathogens. When a virus enters a eukaryotic cell, its gRNA can proliferate indefinitely at the expense of the nucleic acid exchange systems of the cell. Therefore, the main mechanism of protection against coronavirus should be to suppress the gRNA multiplication of the virus and its protein synthesis. To date, the direct pathogenic effect of coronavirus proteins in human cells has not been established. The pathogenic effect of gRNA on cell metabolism may occur. Therefore, specific inhibitors of viral gRNA synthesis are required. Endogenous human piRNAs and miRNAs protecting against pathogenic viruses can be such inhibitors. The piRNAs are actively synthesized in the early stages of embryogenesis and with the development of the organism only in the stem cells of the body [18]. Later, their synthesis decreases and the function of protection against coronaviruses passes to miRNAs synthesized in differentiated somatic cells [18]. We found the 13 piRNAs binding in the 28-nucleotide cluster involve canonical and non-canonical nucleotide pairs complementarily forming bonds while maintaining the double-stranded structure of the piRNA and virus gRNA (Figure 1).

These endogenous piRNAs can be used against coronavirus with the condition that their concentration is limited to avoid causing side effects. Since endogenous piRNAs are abundant, their toxic effect on human target genes is unlikely, but they will protect humans against coronavirus. Schemes of the interaction of piRNAs with the gRNA of the virus are shown in Figure 1. The 13 identified piRNAs of endogenous origin are active antiviral agents, so increasing their concentration in human cells will prevent the development of pathology. Based on these and other piRNAs that bind in the identified cluster, synthetic spiRNAs can be created that will more effectively inhibit the replication of the coronavirus.

### 2.2. Thirty-Nine piRNA Binding Cluster in SARS-CoV-2 gRNA

By analyzing piRNA interactions with SARS-CoV-2 gRNA, we found 39 piRNAs binding in a 31 nt cluster from the 12077 nt (piR-404056) to 12107 nt (piR-396601) site of SARS-CoV-2 gRNA (Table 1). The free binding energy of these piRNAs to the gRNA of the coronavirus varied in ΔG value from −98 kJ/mol to −117 kJ/mol with a mean ΔG value of −106 ± −4 kJ/mol (Table 1).

At an appropriate concentration, the 39 defined piRNAs can affect the multiplication of the coronavirus. Since piRNA synthesis decreases during ontogenesis, the effectiveness of the antiviral activity of these piRNAs will decrease [18]. As the human age increases, the concentration of these piRNAs in human cells will decrease and may become insufficient to suppress viral reproduction, since these piRNAs bind to the mRNAs of their target genes. Creating synthetic spiRNAs based on natural piRNAs would greatly enhance their interaction with the gRNA of the virus, spiRNAs creation should use higher binding energy. Appendix A shows the interaction schemes of some spiRNA with gRNA having a value of −130 kJ/mol or more.

Table 1 shows the quantitative characteristics of the interaction of piRNAs with BSs in the 31 nt gRNA cluster of the SARS-CoV-2 virus. These characteristics show a significantly stronger interaction of spiRNAs compared to endogenous piRNAs. The free binding energy of these spiRNAs to coronavirus gRNA varied from −130 kJ/mol to −142 kJ/mol with a mean ΔG value of −134 ± −5 kJ/mol (Table 1). Based on these data, spiRNAs with the free energy of interaction with viral gRNA of ΔG value of −138 kJ/mol or more can be chosen: piR-188808, piR-188962, piR-189493, piR-189542, piR-189637, piR-190555, piR-190706, piR-190772, piR-192160, piR-194135, piR-194397, piR-392668, piR-396601, piR-401969, piR-404056, piR-406209, and piR-410103.

Next, we evaluated the effect of spiRNA on human mRNA genes. The data obtained indicate that spiRNAs act on human target genes without a significant difference from endogenous piRNAs (Appendix A). Thus, the average value of ΔG is −110 ± −3 kJ/mol. Among the 17,494 human genes studied, no new BSs for spiRNA were found, and therefore, no side effects from the use of spiRNA are expected.

### 2.3. Twenty-Four piRNA Binding Cluster in SARS-CoV-2 gRNA

We identified another cluster of 24 piRNA BSs from 20645 nt (piR-1950681) to 20676 nt (piR-2540716) of 32 nt in the CDS gRNA of coronavirus (Table 2). The free binding energy of these piRNAs to the gRNA of the coronavirus varied from −100 kJ/mol to −127 kJ/mol with a mean ΔG value of −111 ± −7 kJ/mol. The free binding energy of the corresponding spiRNAs to coronavirus gRNA varied from −119 kJ/mol to −151 kJ/mol with a mean ΔG value of −130 ± −8 kJ/mol (Table 2). The most effective inhibitors of coronavirus multiplication are piR-1877632, piR-1930602, piR-1957782, and piR-2526803. Given that piRNAs are on average longer than miRNAs, piRNAs will be more effective in suppressing coronavirus multiplication.

Since RNA polymerase replicates coronavirus gRNA in the cell, piRNA binding at any gRNA site will prevent both protein synthesis and viral gRNA replication. This important circumstance is the basis of the efficiency of coronavirus control with piRNAs.

### 2.4. Twelve piRNA Binding Clusters in SARS-CoV-2 gRNA

In addition to the BS piRNA clusters described above, we identified 12 single piRNAs that most strongly bind to the CDS gRNA of coronavirus (Table 3). The average interaction-free energy of these piRNAs was −139 ± 3 kJ/mol. These piRNAs can be used to create spiRNAs that will bind strongly to coronavirus gRNA (Table 3). Schemes of 12 piRNA and spiRNA interactions with the CDS gRNA SARS-CoV-2 from nt 4670 to nt 29475 are shown in Appendix A. These piRNAs and spiRNAs had a weaker effect on human genes, so no side effects on human genes are expected when these piRNAs and spiRNAs are used in concentrations comparable to endogenous piRNAs. The mean value of the interaction-free energy of these spiRNAs was −170 ± 4 kJ/mol, which ensures the suppression of coronavirus multiplication (Appendix A). Of the 12 spiRNAs, 7 (piR-35553222, piR-703629, piR-1525356, piR-2599982, piR-806264, piR-98504, and piR-3218674) bind in the final gRNA site of the virus with a length of 10%. They will mainly prevent the replication of the RNA of the virus.

## 3. Discussion

We assume that the human body has the ability to counteract SARS-CoV-2 with endogenous piRNAs and miRNAs, and in most cases, the infection is not manifested or is expressed to a weak degree [4]. The strong manifestation of infection is due to abnormalities in the synthesis system of the necessary set of piRNAs and miRNAs. The second aspect of a pronounced manifestation of infection is expressed when abnormalities in the human body contribute to the mechanism of the pathogenic effect of the virus, which is not yet known. Assumptions about the mechanism of pathogenic action of coronavirus have been expressed; however, these assumptions are not reliably established. A distinction should be made between the primary gene response of the organism to coronavirus and the secondary gene response induced by the primary gene response.

The piRNA database contains more than eight million molecules [6,7]. With an average piRNA length of approximately 28 nt, the total length of all molecules is approximately 230 million nucleotides, which is approximately 7% of the entire human genome. The protein-coding part of the human genome is approximately 1.1%, and the total length of genes with the exon-intron organization is 25% [19]. This number of piRNAs seems to be necessary to regulate the expression of protein-coding genes at the translation stage. Considering that the presumed function of piRNAs is to regulate almost all protein-coding genes for coordinated expression of all genes in the genome, this number of these piRNA molecules is quite understandable. In recent years, a large number of publications have appeared showing the involvement of piRNA in a variety of physiological processes and in various diseases [9,13,16,20,21].

Analysis of piRNA BSs in human genes showed that in many protein-coding genes, piRNA BSs are located mainly in the 5’UTR, CDS, and 3’UTR mRNA [21]. In most cases, the BSs are located with overlapping nucleotide sequences in regions called clusters, which are approximately 100 nt long. Typically, such clusters contain dozens of piRNA BSs, indicating a strong dependence of gene expression on piRNAs. Importantly, some clusters contain complexes of multiple piRNA BSs that are present in clusters of different genes. Such piRNA complexes are involved in the regulation of many biological processes, forming a balanced system of cell and organism metabolism regulation.

The presence of piRNA BSs clusters in the SARS-CoV-2 coronavirus genome may be one of the molecular mechanisms protecting the human organism from virus multiplication. The creation of such piRNA complexes counteracting coronaviruses in humans and animals in the course of evolution has taken place over millions of years. This is supported by evidence of different susceptibility of different animals to SARS-CoV-2 [4,20]. At present, due to advances in molecular biology, it is possible to fight successfully against pathogenic viruses with single-stranded RNA genomes. By the example of SARS-CoV-2, piRNAs capable of selectively suppressing these strains can be identified for any strain of coronavirus. This does not require large material costs. Without going into the mechanisms of pathogenic action of SARS-CoV-2 coronavirus, we note that there are assumptions about these mechanisms, and they can be established in the near future.

Analysis of the binding of more than eight million piRNAs to SARS-CoV-2 gRNA showed that more than 100 piRNAs can potentially inhibit viral replication (Table 1, Table 2 and Table 3, Appendix A). The piRNAs can bind in a single gRNA site of a coronavirus encoding a protein or in gRNA sites containing several or dozens of BSs forming clusters of BSs with overlapping nucleotide sequences. Such a number of BSs of human endogenous piRNAs is a natural ability to protect them from coronaviruses. Effective suppression of both coronavirus protein synthesis and coronavirus gRNA replication is a protective reaction developed over millions of years in humans and many animals. All endogenous piRNAs can affect the expression of almost all human genes. Therefore, the problem arises of how to increase the binding of these piRNAs with respect to the gRNA sites of coronavirus. This increase can be obtained by creating spiRNAs in which the nucleotides in the endogenous piRNAs are replaced so that only synonymous A-U and G-C pairs are formed instead of nonsynonymous A-C and G-U pairs. As a result, the difference in the interaction-free energy of spiRNA with coronavirus gRNA increases significantly compared to the difference in the interaction-free energy of target genes and endogenous piRNAs. Since the performed calculations are analyzed under the condition of equal concentrations of gRNA of the virus and target genes mRNA, the real effects of the model experiments should be verified in laboratory experiments. We lack the material resources to conduct such experiments, so we appeal to the scientific community to cooperate to carry out the proposed experiments. The effect of piRNAs on coronavirus can be significantly increased by creating fully complementary spiRNAs based on the discovered endogenous piRNAs with their long nucleotide sequence length (Table 3). The spiRNAs interact significantly more strongly with coronavirus gRNA, and their effect on the translation of human target gene mRNAs can be controlled. It is possible to create synthetic piRNAs or miRNAs capable of having a much stronger effect on coronavirus than on human genes. Combating the constantly detected diversity of SARS-CoV-2 coronavirus strains is easily accomplished by rapid bioinformatic calculations and the speed of obtaining the necessary spiRNAs. Recent work clearly shows that piRNAs in exosomes affect SARS-CoV-2 multiplication [15] and that piRNAs involving PIWI alone can suppress SARS-CoV-2. Unfortunately, studying the role of PIWI in suppressing coronavirus multiplication would be long without establishing to what extent each PIWI can interact with the gRNA of coronavirus strains. Therefore, our proposed methodology will accelerate the development of piRNA-based anti-coronavirus drugs.

## 4. Materials and Methods

The 29903-nucleotide long sequence of the SARS-CoV-2 (NC_045512.2) RNA were downloaded from National Center for Biotechnology Information (NCBI) (http://www.ncbi.nlm.nih.gov, accessed on 5 January 2020). The nucleotide sequences of more than eight million piRNAs were obtained from Wang et al. [7]. The piRNA binding site (BS) in the RNAs were predicted using the MirTarget program [22]. This program predicts the following features of piRNA binding to mRNA: (a) the initiation of piRNA binding to the mRNAs from the first nucleotide of the mRNAs; (b) the localization of the piRNA BSs in the 5′-untranslated region (5′UTR), coding domain sequence (CDS), and 3′-untranslated region (3′UTR) of the mRNAs; (c) the schemes of nucleotide interactions between piRNAs and mRNAs; (d) the free energy of the interaction between piRNA and the mRNA (ΔG, kJ/mol); and the ratio ΔG/ΔGm (%) is determined for each site (ΔGm equals the free energy of piRNA binding with its fully complementary nucleotide sequence). The MirTarget program finds hydrogen bonds between adenine (A) and uracil (U), guanine (G), and cytosine (C), G and U, and A and C. The free energy of interactions (ΔG) of a pair of G and C is equal to 6.37 kJ/mol, a pair of A and U is equal to 4.25 kJ/mol, G and U, A and C equal to 2.12 kJ/mol [23]. The distances between bound A and C (1.04 nm) and G and U (1.02 nm) are similar to the distances between bound G and C and A and U, which are equal to 1.03 nm. These parameters were obtained from published works [24,25,26]. The numbers of hydrogen bonds in the G-C, A-U, G-U, and A-C interactions were 3, 2, 1, and 1, respectively. Consideration of the schemes shows which nucleotides of non-canonical pairs increase the energy of interaction between piRNAs and human mRNA genes [27,28]. A better confirmation of the obtained results is provided by the schemes of the interaction of nucleotides along the entire length of the piRNAs and BSs.

## Figures and Tables

**Figure 1 ijms-23-09919-f001:**
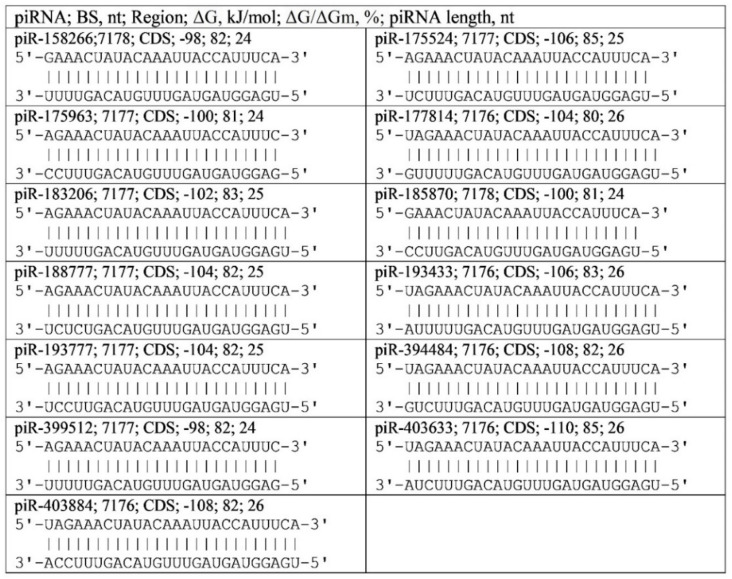
The characteristics of piRNAs and gRNA interactions involving canonical and non-canonical nucleotide pairs. The schemes indicate: the name of the piRNA; the start of the piRNA binding site; the region where the piRNA binding site is located in gRNA; the free energy of interaction of piRNA with gRNA (ΔG); the ratio ΔG/ΔGm (%), where ΔGm equals the free energy of piRNA binding with its fully complementary nucleotide sequence; the length of piRNA.

**Table 1 ijms-23-09919-t001:** Characteristics of the interaction of 39 piRNAs and spiRNAs with the BS RNA cluster of SARS-CoV-2 from 12077 nt to 12107 nt.

piRNA	Start of BSs, nt	ΔG, kJ/mol	ΔG/ΔGm, %	piRNA Length, nt	ΔG*, kJ/mol
piR-404056	12077	−108	81	27	−142
piR-188123	12078	−104	80	26	−138
piR-188808	12078	−108	81	26	−138
piR-188962	12078	−108	82	26	−138
piR-189542	12078	−106	81	26	−138
piR-189637	12078	−106	81	26	−138
piR-190555	12078	−113	84	26	−138
piR-190706	12078	−110	81	26	−138
piR-190772	12078	−104	80	26	−138
piR-192160	12078	−106	81	26	−138
piR-194135	12078	−110	81	26	−138
piR-194397	12078	−108	81	26	−138
piR-392668	12078	−108	82	26	−138
piR-401969	12078	−115	82	26	−138
piR-403862	12078	−104	80	25	−134
piR-406209	12078	−106	81	26	−138
piR-406508	12078	−98	81	24	−127
piR-410103	12078	−108	82	26	−138
piR-187101	12079	−102	81	25	−132
piR-189493	12079	−108	84	25	−132
piR-190719	12079	−106	82	25	−132
piR-191124	12079	−106	82	25	−132
piR-191185	12079	−110	84	25	−132
piR-191492	12079	−106	82	25	−132
piR-192078	12079	−104	80	25	−132
piR-194007	12079	−106	82	25	−132
piR-194172	12079	−104	82	25	−132
piR-403725	12079	−102	81	24	−127
piR-194781	12079	−104	82	25	−132
piR-404008	12079	−104	82	25	−132
piR-405339	12079	−104	82	25	−132
piR-405683	12079	−100	81	24	−127
piR-406684	12079	−102	81	25	−132
piR-180819	12080	−98	81	24	−125
piR-184604	12080	−102	83	24	−125
piR-396601	12081	−117	81	27	−140
piR-360432	12082	−106	81	25	−130
piR-1177268	12082	−102	81	25	−130
piR-345961	12084	−104	82	24	−125

Note. ΔG* is ΔG for spiRNA with ΔG/ΔGm equal to 100%.

**Table 2 ijms-23-09919-t002:** Characteristics of the interaction of 24 piRNAs and spiRNAs with the BS gRNA cluster of SARS-CoV-2 coronavirus from 20645 nt to 20676 nt.

piRNA	Start of BSs, nt	ΔG, kJ/mol	ΔG/ΔGm, %	piRNA Length, nt	ΔG*, kJ/mol
piR-77587	20650	−108	81	25	−127
piR-78640	20650	−106	82	24	−121
piR-89731	20649	−104	82	24	−119
piR-104672	20649	−113	82	26	−132
piR-1874165	20649	−115	84	26	−132
piR-1874441	20648	−110	80	26	−132
piR-1877632	20646	−127	82	29	−151
piR-1901265	20648	−106	81	25	−125
piR-1905481	20650	−110	84	25	−127
piR-1909341	20646	−113	82	26	−132
piR-1916216	20649	−110	81	26	−132
piR-1930602	20648	−115	81	27	−138
piR-1950681	20645	−119	81	27	−136
piR-1955234	20647	−106	81	25	−125
piR-1957782	20646	−123	81	29	−151
piR-1968262	20647	−110	80	26	−132
piR-1970873	20646	−108	85	24	−121
piR-1978038	20645	−100	81	24	−119
piR-1981597	20645	−108	82	25	−125
piR-2526803	20647	−125	83	28	−144
piR-2540472	20647	−104	83	24	−121
piR-2540716	20650	−110	81	26	−132
piR-2557847	20645	−113	82	26	−132
piR-2574698	20645	−108	81	25	−125

Note. ΔG* is ΔG for spiRNA with ΔG/ΔGm equal to 100%.

**Table 3 ijms-23-09919-t003:** Characteristics of 12 piRNAs and corresponding spiRNAs effectively interacting with the gRNA of SARS-CoV-2.

piRNA	Start of BSs, nt	ΔG, kJ/mol	ΔG/ΔGm, %	Length, nt	ΔG*, kJ/mol
piR-2047904	4670	−136	81	32	−168
piR-912075	9102	−140	80	33	−175
piR-2352720	9123	−138	81	34	−170
piR-2490582	10012	−142	83	33	−171
piR-1491787	17115	−138	80	32	−173
piR-3555322	21712	−136	82	34	−166
piR-703629	25672	−142	82	31	−173
piR-1525356	26106	−136	80	32	−170
piR-2599982	27070	−144	82	34	−176
piR-806264	28427	−142	83	33	−171
piR-98504	29024	−136	84	32	−162
piR-3218674	29475	−138	81	34	−170

Note. ΔG* is ΔG for spiRNA with ΔG/ΔGm equal to 100%.

## Data Availability

Not applicable.

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
