# Peer review of "In Silico Study of piRNA Interactions with the SARS-CoV-2 Genome"

_ijms, 2022, doi:10.3390/ijms23179919_

Round 1

Reviewer 1 Report

The authors present an analysis of the binding of piRNAs to the coronavirus genome.  While there are some interactions discovered from the large database of piRNAs studied, these interactions are mostly short and not very strong interactions.  That being said the study is scientifically valid and I think it is worthy of publication if the authors address a few comments (not listed in order of importance)

1. The free energies in the introduction are listed as positive, so this should be switched to match the remainder of the document.

2. Free energies throughout are given in kJ/mol, but it is far more common in this area to discuss free energies in kcal/mol, so I think it would be best for the reader if these were converted.

3. Given that none of the interactions seem to cover ribosome or polymerase binding sites, in order for the piRNAs to inhibit the virus, they would need to cause ribosome or polymerase unbinding, while translation or replication is ongoing.  This should be discussed, with reference to the interaction energies of the piRNAs.  

4. Coronavirus replicates using a -ssRNA template and translates via subgenomic fragments of the +ssRNA - how would this affect the effectiveness of the piRNA?  Did the authors scan against the -ssRNA template as well?  Given the different concentrations of these RNAs in the cell, this may have an effect.

If these comments are address I think it is suitable for publication.   

Reviewer 2 Report

This original work highlights important problem of new approaches for coronavirus control by microRNA. piRNA (Piwi interacting RNA) were not studied from this point of view. Thus, this paper has significant novelty passing way to new drug target search to control COVID at molecular level.

The authors used novel software and comprehensive database search to model potential human piRNA interaction in the coronavirus genome.  It is necessary to underline that it is computer testing, a model, not yet wet-lab experiments (from the Abstract).

Line 18: “We investigated the effects..” - change to

“We modeled the effects..” or “We estimated the effects in silico..” - to show that is computational test.

Line 25: “have little effect on..” - change to “have little computationally predicted effect” or “The computer model shows that ...piRNAs have little effect...” - rephrase to show that it is not wet-lab test yet.

The findings could be discussed in more details. The hypothesis is interesting, but the findings need some explanation, at least in the discussion (the oligopeptides in binding regions - LETIQITIS and others. What is known about the protein, elements of protein secondary structure, active sites, how conservative are these regions. Might be interacting and controlling effects ion other organisms? Even if no known information it is worthy to discuss).

The manuscript should be updated.

I suggest even update the title - “In Silico Interaction piRNA With Sars-CoV-2 Genome” - the wording is incomplete.

It could be to “In Silico study of piRNA Interactions with Sars-CoV-2 Genome” or “In silico Analysis of Binding piRNA in Sars-CoV-2 Genome”. Rephrase to add words ‘model’, ‘investigation’, or ‘analysis’

Remarks:

‘PIWI’ could be written as ‘PiWi” or ‘Piwi’, not in all capital letters.

Line 29: Keywords. Add keyword  ‘coronavirus genome’. Keyword ‘pandemic’ may to remove.

Keyword ‘treatment’ is better change to ‘drug target’. The paper is not about treatment yet.

Line 34 and 36: “coronavirus gRNA”. It is redundant abbreviation here. Whole coronavirus genome is RNA.  My rephrase and not use ‘gRNA’.

Line 40: “were successful in experiments” - need give references for these experiments.

Line 41: “in cell culture and hamsters” - it is about different experiments? Rephrase the sentence here. Add word ‘human’ - “in human cell culture”, provide the reference. Cite test on hamsters separately.

Line 43: ‘ the drugs” - name these drugs. Or write “miRNA-based drugs” to show that is had limitations.

Line 51: “opened up new possibilities for fighting...” - too strong statement. Write “may open up...” or “open potential ways for ...”

 Line 61: “[11-15]” - bulk citation (5 references together). Give reference to these reports separately, try to rephrase.

Line 62: “are known [16]” - it is better write as “were reviewed recently in [16]”. Or give several references for these reports.

Line 72: “long sequences” - assume it is one sequence, not sequenceS? Or the authors used different variants?

Line 74: NCBI web-link is too common. Give more precise link to the sequence archive.

Line 75: “BSs” - give abbreviation for BS in the text again (it was only in the Abstract).

“mRNAs of genes” - may remove this wording. It is about binding everywhere in genome, not only in genes.

Line 83: ‘program finds’ - change ‘finds’ to ‘predicts’. It is computational prediction, not finding in experimental data.

Next paragraph (lines 85-88) with the references [21-23] could be commented like “The program used the bonds parameters described in [21-23]”. Otherwise it is not clear it these parameters are unique only for this study, or how it is related to the topic of the paper.

Line 92: “A better confirmation of the obtained results than “wet” experiments...”  - how it is possible to make better than ‘wet’ experiments? Please add the reference, that is was shown, or rephrase that “predicted results fit to ‘wet-lab’ experimental data ... as it was shown in ...”

Line 102: “We can evaluate the side effects of” - add word ‘computationally’  or ‘in silico’ when describing the side effects.. It is too strong statement.

Line 108: “many piRNAs” - write how many - give the number. Does Table 1 has all the found piRNA bindings?

Line 116: “sequences. human” - typo (extra dot or word ‘human’).

Line 118: “from 7176 nt (piR-177814) to 7178  nt” - Staring position only are given in the text .

Should write “by starting position”, rephrase, indicate total region location [7176;7176+28].

May update the figure and make some graphics of overlaps to show how these 24-25 nt long RNA bind to genomic region of total length 28 nt (by line graphics, not by text)

Figure 1 has genomic positions on top of the alignments (from 5’)?

Assume same positions are in top lines in the figure, and piRNA may differ?

Line 120: “LETIQITIS oligopeptide in the ORF1ab protein” - need comment about this peptide, and ORF1a protein. What is known about its function?

If it just translation or this amino acid sequence has some known structure of function?

Line 121: “reading frame”. What is about piRNA binding found ? Is it in 0,+1... reading frame for the LETIQITIS oligopeptide?

Figure 1 should be commented. Usually sequences are shown from left to right (from 5’ to 3’)

Here on top we see position 7178, then 7177, then 7176

Line 127: “. Only such large sets of piRNAs” - remove word ‘only’

Line 132: “effect of coronavirus proteins... not been established” -may add a reference, or write word ‘direct effect’

Line 133: “gRNA on cell metabolism may occur” - could add a reference for such a hypothesis? Or add details. Assume if RNA is more stable than protein, effects of coronavirus RNA are more probable than effects of coronavirus proteins.

Line 137: “only in the stem cells of the body.”  - it is common statement, but may add a reference.

Line 139: “The 13 piRNAs ... involved” - add “We found...”

change ‘involved’ to ‘involve’, to make more mild statement about computational prediction.

Line 152: Figure legend. Add comment what is shown in the schemes  - for example, “The genome pRNA name, genome position, binding energy and piRNA length are shown in top line, then genomic sequence (from 5’- to 3’-end) and aligned piRNA sequence (from 3’)”

It is good to show how LETIQITIS oligopeptide is located relative to this binding locus.

Line 158: “RATLQAIASEF oligopeptide of the ORF1ab protein” - it is good to comment, what is known about this peptide and ORF1ab protein. Does it has crucial function for coronavirus?

Table 1. It has another presentation of binding. Positions are from 12077 nt to 12107 nt in the reference genome.

But the lines in the Table are not sorted (not by position, not by energy. Logically it is good to sort lines by the positions, then we will see some order, it will resemble presentation in Figure 1 by other way

Line 171: “Creating synthetic spiRNAs...” then “Figure S1..” It is better add a phrase in between “But spiRNAs creation should use higher binding energy. Thus, we listed the details for such binding in Figure S1.

Then, add “Figure S1 resemble Figure 1 style”.

Line 184: “almost equally” - should be statistical meaning in the phrase like “no significant difference was found”.

Line 196: “PKLQSSQAWQP of the ORF1ab protein” - need comment on the knowm function of the protein here too.

Otherwise, it is not clear why the amino acid sequences were shown.

Line 223: “Based on the obtained results,” - I’d remove this wording. These computational results not are the base for the assumption. IT could be referred to some previous observation[4] if need to cite.

Line 233: “The piRNA database contains more than eight million molecules [6-8].” - not need 3 references to one database. Add instead data on experimental testing of miRNA in human by sequencing.

Line 282: “tens of thousands of Earth's inhabitants are dying every day” - it is true, unfortunately, but such phrases are not in science paper style. Please try to concentrate on the paper topic.

Line 285: “(Table 4)” - there are only Tables 1-3 in the text.

Ref. 2 has a typo: “doi: https://doi”

The refs. 7 and 8 in the reference list are duplication.
